# Molecular and Cellular Mechanisms Associated with Effects of Molecular Hydrogen in Cardiovascular and Central Nervous Systems

**DOI:** 10.3390/antiox9121281

**Published:** 2020-12-15

**Authors:** Miroslav Barancik, Branislav Kura, Tyler W. LeBaron, Roberto Bolli, Jozef Buday, Jan Slezak

**Affiliations:** 1Centre of Experimental Medicine, Slovak Academy of Sciences, 84104 Bratislava, Slovakia; miroslav.barancik@savba.sk (M.B.); branislav.kura@savba.sk (B.K.); lebaront@molecularhydrogeninstitute.com (T.W.L.); 2Faculty of Medicine, Institute of Physiology, Comenius University in Bratislava, 84215 Bratislava, Slovakia; 3Molecular Hydrogen Institute, Enoch, UT 84721, USA; 4Department of Kinesiology and Outdoor Recreation, Southern Utah University, Cedar City, UT 84720, USA; 5Department of Medicine, Institute of Molecular Cardiology, University of Louisville, Louisville, KY 40292, USA; roberto.bolli@louisville.edu; 6Department of Psychiatry, First Faculty of Medicine, Charles University in Prague and General University Hospital in Prague, 12108 Prague, Czech Republic; Jozef.Buday@vfn.cz

**Keywords:** molecular hydrogen, oxidative stress, autophagy, matrix metalloproteinases

## Abstract

The increased production of reactive oxygen species and oxidative stress are important factors contributing to the development of diseases of the cardiovascular and central nervous systems. Molecular hydrogen is recognized as an emerging therapeutic, and its positive effects in the treatment of pathologies have been documented in both experimental and clinical studies. The therapeutic potential of hydrogen is attributed to several major molecular mechanisms. This review focuses on the effects of hydrogen on the cardiovascular and central nervous systems, and summarizes current knowledge about its actions, including the regulation of redox and intracellular signaling, alterations in gene expressions, and modulation of cellular responses (e.g., autophagy, apoptosis, and tissue remodeling). We summarize the functions of hydrogen as a regulator of nuclear factor erythroid 2-related factor 2 (Nrf2)-mediated redox signaling and the association of hydrogen with mitochondria as an important target of its therapeutic action. The antioxidant functions of hydrogen are closely associated with protein kinase signaling pathways, and we discuss possible roles of the phosphoinositide 3-kinase/protein kinase B (PI3K/Akt) and Wnt/β-catenin pathways, which are mediated through glycogen synthase kinase 3β and its involvement in the regulation of cellular apoptosis. Additionally, current knowledge about the role of molecular hydrogen in the modulation of autophagy and matrix metalloproteinases-mediated tissue remodeling, which are other responses to cellular stress, is summarized in this review.

## 1. Introduction

### 1.1. Molecular Hydrogen and Its Potential Use in Therapy

Hydrogen is a colorless and odorless, diatomic gas, which, in mammals, is produced by intestinal bacteria. The hydrogen molecule is small (molecular weight 2 Da), electrically neutral, and nonpolar. These properties allow for its easy entrance into cells and rapid diffusion across all biological cell membranes. In this way, molecular hydrogen can reach the subcellular compartments, such as mitochondria and endo/sarcoplasmic reticulum, and nuclei, which are the primary sites of reactive oxygen species (ROS) generation and DNA damage, respectively. Moreover, it can easily penetrate several barriers, such as the blood-brain barrier, the placental barrier, and the testis barrier.

Molecular hydrogen is currently recognized as an emerging therapeutic, as its supplemental application exerts protective effects in cardiovascular diseases [1,2], neurodegenerative diseases [3], inflammatory diseases [4], neuromuscular disorders [5], metabolic syndrome [6] diabetes [7,8], kidney disorders [9,10], and cancer [11]. The protective effects of molecular hydrogen are largely related to its antiapoptotic, anti-inflammatory, and antioxidative actions (Figure 1).

Molecular hydrogen has no known negative side effects on cells. Its use does not disturb cellular metabolic redox reactions, intracellular signaling (e.g., the signaling role of reactive oxygen species—ROS) [12], or physiological metabolic and enzymatic reactions. Hydrogen has very low reactivity with other gases at therapeutic concentrations and lacks reactivity to nitric oxide (NO^•^). This allows its administration with other therapeutic gases, including inhaled anesthetic agents, and enables the concomitant administration of hydrogen with NO^•^. The administration of molecular hydrogen can be performed in several ways. The most common methods are inhalation of molecular hydrogen as a gas [13], application of a hydrogen-rich solution [3], or administration of hydrogen-loaded eye drops [14]. Another method is the use of hydrogen-rich water, which is more convenient for long-term treatment.

The positive effects of molecular hydrogen have been demonstrated not only in animal experiments, but also in clinical trials. In a single-center prospective, open-label, blinded study, Katsumata et al. [15] investigated the feasibility and effects of hydrogen inhalation on infarct size and adverse left ventricular remodeling after a primary percutaneous coronary intervention (PCI) for ST-elevated myocardial infarction (STEMI). They found that inhalation of 1.3% H_2_ during PCI resulted in left ventricular (LV) reverse remodeling at six months after STEMI. The positive effects of H_2_ were also demonstrated in metabolic syndrome (*n* = 60) using a double-blinded, placebo-controlled trial [6]. The consumption of H_2_-rich water for 24 weeks led to a significant reduction in blood cholesterol, glucose levels, attenuated serum hemoglobin A1c, and improved biomarkers of inflammation and redox homeostasis as compared to the placebo group. Similarly, an earlier study by Kajiyama et al. [16] reported that drinking H_2_-rich water for eight weeks (with a 12-week washout period) significantly decreased the levels of modified low-density lipoprotein (LDL) cholesterol, small dense LDL, urinary 8-isoprostanes, serum concentrations of oxidized LDL and free fatty acids, and increased plasma levels of adiponectin and extracellular-superoxide dismutase in patients with diabetes mellitus type 2. After H_2_ therapy, normalization of the oral glucose tolerance test occurred in four out of six patients with impaired glucose tolerance. In another randomized, double-blinded, placebo-controlled trial, the efficacy of drinking of hydrogen water for 48 weeks in Japanese patients with levodopa-medicated Parkinson’s disease (PD) was investigated [17]. Despite the small number of patients and the short duration of the trial, the results clearly demonstrated the beneficial effects of hydrogen water. Drinking hydrogen water was shown to be safe and well tolerated, and was associated with a significant improvement in Total Unified Parkinson’s Disease Rating Scale (UPDRS) scores for patients with PD. Sakai et al. [18] showed that molecular hydrogen can be a useful modulator of blood vessel function. The data observed in their study documented that the vasculature of volunteers drinking daily water containing a high concentration of hydrogen was protective against shear stress-derived detrimental ROS. The protective effects of hydrogen may have been mediated by a reduction of detrimental ROS, and were associated with preserving the bioavailability of nitric oxide (NO) and maintaining the NO-mediated vasomotor response.

Importantly, from the perspective of clinical use, the administration of molecular hydrogen is safe and is not associated with undesirable toxic effects.

### 1.2. Molecular Hydrogen and The Cardiovascular System

Diseases of the cardiovascular system are among the most serious medical problems, and represent a major cause of health complications and morbidity in modern society [19]. Increased production of ROS and oxidative stress are important factors that contribute to the development of cardiovascular diseases (CVD) such as hypertension [20], cardiac hypertrophy [21,22], and heart failure [23]. A very important and preventable cause of CVD is hypertension, which may, without appropriate treatment, lead to cardiac remodeling followed by left ventricular hypertrophy, and potentially also heart failure [24].

Ischemia/reperfusion (I/R) injury also plays an important role in the induction of cardiac remodeling. Reperfusion is induced by the blood supply returning to the heart after a period of ischemia, and is associated with the induction of oxidative stress injury, calcium overload, inflammation, and apoptosis [25,26,27,28]. This can impair cardiac function and lead to myocardial infarction and malignant arrhythmias.

Various studies have employed several potential strategies for the prevention, control, and treatment of cardiovascular diseases including the reduction of increased ROS production and oxidative stress, as well as targeting the signaling pathways modulated by ROS [29,30,31,32]. The antioxidative, anti-inflammatory, and antiapoptotic properties of molecular hydrogen may explain why its clinical application may result in the improvement of oxidative stress-related cardiovascular diseases.

The positive effects of molecular hydrogen on diseases related to the cardiovascular system have been reported in several studies. Hydrogen inhalation significantly improved cardiac and brain function in a rat model of cardiac arrest [1], and chronic treatment of spontaneous hypertensive rats with hydrogen-rich saline (HRS) attenuated left ventricular hypertrophy development in these animals [33]. The protective effects of hydrogen on left ventricular function were also observed in other studies demonstrating its ability to attenuate left ventricular remodeling induced by intermittent hypoxia [34] or ischemia/reperfusion (I/R) injury [13]. The positive role of molecular hydrogen treatment in modulating myocardial responses to ischemia/reperfusion injury has been demonstrated in several other studies where different methods of hydrogen application, such as inhalation of hydrogen gas [13] or intraperitoneal application of hydrogen-rich saline, were used [35]. Inhalation of hydrogen gas during reperfusion reduced infarct size in models of cardiac I/R injury in rats [13] as well as in dogs [36]. In the canine model, it was suggested that the cardioprotective effects of hydrogen were realized via the opening of mitochondrial ATP-sensitive potassium channels (mitK-ATP) and the subsequent inhibition of mitochondrial permeability transition pores [36]. mitK(ATP) channels were found to be involved in myocardial responses to ischemia/repefusion [37], and pretreatment with diazoxide, mitK(ATP) channel opener, was found to protect rat heart against ischemia/reperfusion injury [38]. An investigation of the in vivo effects of hydrogen on myocardial I/R injury in rats found that intraperitoneal application of HRS reduced infarct size and also improved cardiac dysfunction. I/R injury caused excessive release of pro-inflammatory molecules (TNF-α, IL-1β, IL-6, and HMGB1), and the hydrogen-mediated cardioprotection was associated with a reduction of these I/R-induced inflammatory responses in myocardial tissue.

Another study demonstrated that molecular hydrogen potentiates the beneficial infarct-sparing effect of hypoxic postconditioning (HPostC) in isolated rat hearts [2]. The infusion of Krebs-Henseleit buffer with molecular hydrogen during HPostC further decreased infarct size, attenuated severe arrhythmias, and significantly restored heart function compared with HPostC alone. Interestingly, one group [39] found that hydrogen gas can attenuate myocardial I/R injury in rats, independent of postconditioning. Compared with ischemic postconditioning, hydrogen had a better protective effect on I/R injury; this was associated with the attenuation of endoplasmic reticulum stress and the downregulation of excessive autophagy [39]. Treatment with HRS was also found to attenuate the myocardial injury and apoptosis in heart tissue induced by a cardiopulmonary bypass (CPB). The available evidence indicated that the protective effects of HRS involved opposite effects on two distinct signaling pathways, i.e., attenuation of the PI3K/Akt pathway [40] and upregulation of JAK2/STAT3 signaling [41].

### 1.3. Molecular Hydrogen and the Central Nervous System

The nonpolar nature and low molecular weight (2 Da) of molecular hydrogen allow it to easily penetrate biological membranes. This is important in the central nervous system (CNS), because hydrogen can penetrate the membranes that make up the blood-brain barrier (BBB), which plays a pivotal role in the protection of the CNS. Several lines of evidence point to oxidative stress, the activation of matrix metalloproteinases (MMPs), and inflammation as mechanisms linking some pathological conditions, such as cardiovascular diseases and hypertension, to BBB breakdown [42,43].

Vital to the regulation of BBB permeability is the integrity of the endothelial cells. Disruption of this integrity can lead to a dysfunction of the BBB, which can cause neurological disorders such as brain injury and neurodegenerative disorders, and may play a significant role in the pathogenesis of vascular dementia [44,45]. Disruption of BBB function is followed by blood-to-brain extravasation of circulating neuro-inflammatory molecules, which may increase the risk of brain injury. Some cytokines and chemokines, such as IL-6 and TNF-α, are known to be transported across the BBB from the blood into the brain [46]. Moreover, some studies have shown that circulating peripheral immune cells, i.e., macrophages, invade the CNS [47,48]. The crosstalk between the signaling cascades underlying oxidative stress and the inflammatory responses may be a critical factor in neurodegenerative disorders [49,50].

The penetration of molecular hydrogen through the membranes of the BBB and its unrestricted access to the CNS are unique characteristics, shared by only a few therapeutics. It has been found that hydrogen gas inhalation suppressed redox stress and BBB disruption by reducing mast cell activation and degranulation [51]. Moreover, the observed effects of hydrogen were also associated with the suppression of brain edema and neurological deficits [51]. HRS was found to ameliorate brain edema and decrease infarct volume also in the neonatal brain injury in mice. Other studies have demonstrated that supplemental molecular hydrogen improved clinical features in neuromuscular and neurodegenerative diseases [17,52].

The protective effects of molecular hydrogen in the central nervous system are related to the modulation of cellular responses to stress conditions, and are realized through several cellular mechanisms. In 2007, Ohsawa et al. [53] reported that gaseous molecular hydrogen acts as a therapeutic and preventive antioxidant by selectively reducing the levels of strong oxidants, such as the hydroxyl radical (^•^OH) and peroxynitrite (ONOO–), in cells [53]. The protective effects of molecular hydrogen resulted in the suppression of I/R injury in the brain. Molecular hydrogen selectively reduces the levels of the highly toxic hydroxyl radicals and peroxynitrite, but not superoxide, hydrogen peroxide, or nitric oxide [53].

When contemplating the mechanisms for the anti-inflammatory effects of molecular hydrogen in the brain, both the neuro-immunological interactions and crosstalk with oxidative stress need to be considered. Important hydrogen-mediated protective effects include the buffering of oxidative stress, downregulation of endoplasmic reticulum (ER), stress, inhibition of apoptosis, suppression of inflammatory responses, and regulation of autophagy machinery.

## 2. Mechanisms and Cellular Systems Involved in the Actions of Molecular Hydrogen

The effects of molecular hydrogen on various diseases may be attributed to several molecular mechanisms. Hydrogen was first reported to be a selective scavenger of ^•^OH and peroxynitrite [53]; these reactive molecules are primary direct targets of hydrogen. However, mounting evidence suggests that hydrogen can also function as a signal modulator [54,55,56], and several molecules are mediators that are secondarily changed by the administration of hydrogen. The radical scavenging and signal modulating activities of molecular hydrogen are closely related to modulation (regulation) of redox signaling and alterations in gene expressions [54]. Next, we will focus on the role of molecular hydrogen in modulating redox status as well as intracellular protein signaling and the consequences thereof on gene expression, autophagy, and matrix metalloproteinases.

### 2.1. Molecular Hydrogen as Regulator of Redox Signaling

Molecular hydrogen has been reported to be an antioxidant that protects cells against oxidative stress through selectively decreasing cellular levels of hydroxyl radicals (^•^OH) and peroxynitrite (ONOO−) [53]. The stoichiometric reaction between H_2_ and hydroxyl radicals is:H_2_ + 2^•^OH => 2 H_2_O 

Although hydrogen cannot eliminate peroxynitrite as efficiently as hydroxyl radicals, in rodents, it was found that hydrogen can efficiently reduce nitro-tyrosine formation, which is induced by nitric oxide (NO^•^) via the production of peroxynitrite [57,58]. NO^•^ is a gaseous molecule that also exerts therapeutic effects including relaxation of blood vessels and inhibition of platelet aggregation [59]. NO^•^, however, can be toxic at higher concentrations because it leads to peroxynitrite-mediated production of nitrotyrosine, which compromises protein functions. Part of the effects of hydrogen may thus be attributed to the reduced production of nitrotyrosine [58].

Molecular hydrogen reduces oxidative stress not only directly, but also indirectly by inducing antioxidation systems, including heme oxygenase-1 (HO-1) [60,61], superoxide dismutase (SOD) [7,9], catalase [62], and myeloperoxidase (Figure 1) [62,63]. In a rat model of traumatic brain injury, it was observed that the beneficial effects of hydrogen inhalation were mediated by the reduction of oxidative stress and the stimulation of enzymatic activities of the endogenous antioxidants SOD and catalase [64]. Beneficial effects of hydrogen on the activities of antioxidant enzymes were also observed by Guan et al. [9], who discovered that molecular hydrogen protected against renal dysfunction induced by chronic intermittent hypoxia. Hydrogen was shown to alleviate oxidative damage by promoting the upregulation of SOD and glutathione peroxidase (GSH-Px) activities and increasing the GSH/GSSG ratio. The effects of hydrogen were also associated with a reduction of malondialdehyde (MDA) content (a product of oxidative stress). Other studies support the antioxidant functions of molecular hydrogen through the Nrf2/ARE [54,65,66] pathway. This pathway plays an essential role in protection against oxidative stress and in the transcriptional regulation of numerous other antioxidant and cytoprotective proteins [49]. Nrf2 (nuclear factor erythroid 2-related factor 2) is a transcription factor playing an important role in the redox-sensitive regulation of the expression of several endogenous antioxidants and detoxification enzymes [67,68]. Under normal conditions, Nrf2 is inhibited by Kelch-like ECH-associated protein 1 (Keap1), which mediates the Cullin3/Rbx1-dependent polyubiquitination of Nrf2 and its subsequent proteosomal degradation [69]. After exposure of cells to stress, the cysteine residues of Keap1 are modified by oxidative/electrophilic molecules, and Nrf2 escapes Keap1-mediated repression. When Nrf2 is not ubiquitinated, it translocates in the nucleus, where it forms heterodimers with small MAF or JUN proteins and binds to the antioxidant response element (ARE), i.e., the upstream promoter region of many antioxidative genes, and initiates their transcription [19,70]. Regulation of the Nrf2/ARE pathway is generally dependent on the duration and intensity of the oxidative stress. The above mentioned effects are realized above all during acute stress. Prolonged stress induces negative modulation or downregulation of Nrf2 activity, and decreases or arrests antioxidant and detoxification responses. Glycogen synthase kinase 3β (GSK-3β) plays an important role in this modulation by phosphorylating threonine residues of Fyn kinases. The Fyn kinase then translocates into the nucleus where it phosphorylates Nrf2, which leads to Nrf2 export out of the nucleus to the cytoplasm, where it is exposed to ubiquitination and proteasome degradation [71].

The role of Nrf2 in mediating the effects of hydrogen is supported by the results of a study showing that hydrogen gas reduced hyperoxic lung injury via the Nrf2 pathway, and through the induction of Nrf2-dependent genes, such as HO-1 [66]. It was also demonstrated that hydrogen can play a significant antioxidative role in the brain after focal cerebral ischemia reperfusion through upregulation of HO-1 levels [72]. Moreover, recent data indicated that the Nrf2/ARE signaling pathway and upregulation of HO-1 are involved in the neuroprotective effects of hydrogen-rich saline in mice with experimental autoimmune encephalomyelitis [73].

### 2.2. Molecular Hydrogen and Mitochondria

Mitochondria are important organelles involved in several essential cellular functions, such as energy production (ATP), cell differentiation, regulation of calcium homeostasis, and signal transduction [74,75,76]. They also significantly contribute to cellular stress responses associated with cell death. The mitochondria-mediated regulation of apoptosis [77,78] and autophagy [79] is an important biological process. Mitochondrial dysfunction contributes to various human diseases. Mitochondria are known as a major sources of cellular ROS production. The process of oxidative phosphorylation leads to the conversion of oxygen (O_2_) into water (H_2_O) by four-electron reduction. However, a small percentage of O_2_ is converted into superoxide anion radicals by one-electron reduction. Superoxide is decomposed with mitochondrial superoxide dismutase (SOD) and converted to O_2_ and hydrogen peroxide (H_2_O_2_).

The physical properties of molecular hydrogen allow its effective penetration through subcellular compartments, such as mitochondria [80]. Mitochondria are an important therapeutic target, and so the small hydrogen molecule could be applicable for interventions in diseases related to mitochondria. The potential effects of molecular hydrogen have been investigated in several studies. In one model of isolated mitochondria, it was found that molecular hydrogen could suppress superoxide generation in complex I [81]. The same authors demonstrated that the presence of molecular hydrogen in culture media reduced the membrane potential in living human lung cells (A549) [81]. Based on the results of both in vitro and in vivo studies, the authors assumed that electrons released by molecular hydrogen could be donated to the iron-sulfur N2 cluster in complex I of the respiratory chain. In this way, H_2_ may trigger a conformational change in this complex and affect transmembrane proton transfer and/or uncoupling of the membrane potential. Therefore, it was proposed that H_2_ may function as a rectifier of mitochondrial electron flow in the disordered or pathological states when the accumulation of electrons leads to ROS production [82].

Positive effects of molecular hydrogen on mitochondria through the activation of mitochondrial unfolded protein response (mtUPR) pathways have also been demonstrated. mtUPR is a defense mechanism that is activated by stress occurring in the mitochondrial matrix, when damaged proteins accumulate in the mitochondrial matrix and exceed the maximal capacity of the protein folding apparatus [83]. It was found that molecular hydrogen induced this mitochondrial defense mechanism by induction of mtUPR-related proteins expression and via histone 3 (H3) demethylase induction and modification of H3 methylation on lysine 27 (H3K27) [66,84]. The positive effects of hydrogen were also documented by Iuchi et al. [85], who found that molecular hydrogen can suppress tert-butyl hydroperoxide-induced cell death by reducing mitochondrial dysfunction and fatty acid peroxidation [85].

These possible mechanisms involved in the effects of molecular hydrogen may explain the results in recent studies showing protective effects of hydrogen-rich saline against experimental diabetic peripheral neuropathy in rats, which was associated with the activation of mitochondrial ATP-sensitive potassium channels [86]. Moreover, the application of 5-hydroxydecanoate, a mitochondrial ATP-sensitive potassium channels inhibitor, eliminated the neuroprotective effects of hydrogen-rich saline treatment. ATP-sensitive potassium channels can be found in the plasma membrane and the inner membrane of mitochondria [87]. It was demonstrated that these mitochondrial channels play an important role in the protection of myocardial cells against injuries [88], and that their activation can inhibit the apoptosis induced by hydrogen peroxide [89]. Nrf2 is an important regulator of redox signaling. A recent study showed that hydrogen-rich saline can alleviate mitochondrial dysfunction via the Nrf2 pathway [90]. The authors found that sepsis-associated encephalopathy (SAE) led to mitochondrial dysfunction. Hydrogen-rich saline improved the function of mitochondria, as demonstrated by an increase of the mitochondrial membrane potential (MMP), respiratory control ratio (RCR), and ATP release. In addition, hydrogen-rich saline alleviated SAE-induced changes and the production of ROS. The relationship between hydrogen and the Nrf2 pathway was confirmed by findings that the protective effects of hydrogen occurred in wild type but not Nrf2-knockout mice.

Gvozdjakova et al. [91] demonstrated that molecular hydrogen stimulates myocardial mitochondrial function in rats. Drinking molecular hydrogen-rich water (HRW) resulted in increased levels of ATP production at Complexes I and II in the rat cardiac mitochondria. Similarly, coenzyme Q9 levels in the rat plasma, myocardial tissue, and mitochondria were increased after HRW administration.

### 2.3. Molecular Hydrogen and Modulation of Intracellular Protein Kinases Signal Transduction

Molecular hydrogen plays a role not only as a potential free radical scavenger during oxidative stress, but it can also act as a modulator of intracellular signaling mediated by protein kinases. This signaling role is facilitated by the physical properties of molecular hydrogen, namely, the fact that it can easily diffuse throughout tissues and cells. In this way, hydrogen represents a gaseous-signaling molecule, similar to NO^•^. Importantly, molecular hydrogen can reduce oxidative stress, but it cannot directly eliminate functionally important signaling ROS [12,53]. Several studies have documented that molecular hydrogen may exert its effects by targeting several protein kinases [9,55,92,93,94]. The effects of hydrogen on signaling are not unidirectional, and both stimulatory and inhibitory roles of molecular hydrogen in the activation of distinct protein kinase cascades have been demonstrated [54].

The cellular effects of molecular hydrogen seem to be mediated by several intracellular protein kinase pathways. We will focus on the PI3K/Akt and Wnt/β-catenin (Wingless-type mouse mammary tumor virus integration site family member/beta catenin) pathways, which are associated with glycogen synthase kinase 3β (GSK3β), an enzyme that plays an important role in the regulation of cellular endogenous apoptosis.

#### 2.3.1. Molecular Hydrogen and the PI3K/Akt/GSK-3β Signaling Pathway

The PI3K/Akt kinase signaling pathway is a key player in the regulation of the protective induction of Nrf2/ARE during acute oxidative stress. However, in long-term stress, Nrf2 is deactivated to various degrees by kinase cascades associated with the activation of glycogen synthase kinase-3β (GSK3β), as explained above via phosphorylating Fyn kinase, which results in the attenuation or complete cessation of the antioxidant and detoxication response [19,95]. GSK3β is a serine/threonine kinase and the major downstream target of the PI3K/Akt pathway. This enzyme is phosphorylated by Akt kinase and can directly phosphorylate Nrf2 [96]. GSK3β also regulates the cellular endogenous apoptosis pathway, and its downstream target MCL-1 (myeloid lymphoma 1) in the Bcl-2 family [97]. The PI3K/Akt/GSK3β signaling pathway mediates cell survival and plays an important role in I/R injury in the heart [98], brain [99], and kidney [100]. Moreover, this pathway is involved in regulating the function of cerebral blood vessels, and inhibiting its function may aggravate neuronal injury. GSK-3β is considered a therapeutic target for a variety of nervous system diseases. Several studies have indicated that GSK-3β may regulate neuronal apoptosis through the activation of NF-κB or β-catenin signaling pathways [101,102,103].

Several lines of evidence indicate that at least part of the protective effects of molecular hydrogen are related to the regulation of the PI3K/Akt/GSK3β signaling pathway. HRS was found to protect cerebral microvascular endothelial cells from apoptosis after hypoxia/reoxygenation by inhibiting the PI3K/Akt/GSK3β pathway [104]. HRS was also protective against brain injury induced by cardiopulmonary bypass; this protection was associated with the inhibition of apoptosis through the PI3K/Akt/GSK3β pathway [104]. This inhibition of apoptosis by HRS was realized through downregulation of Akt and GSK3β activity, which are core components of this pathway, and through the inhibition of proapoptotic caspase-3 and Bax expression levels. Inactivation of the Akt kinase pathway also played a role in HRS-induced attenuation of vascular smooth muscle cell proliferation and neointimal hyperplasia. In addition to the Akt kinase pathway, the effects of HRS were also associated with inhibition of ROS production and inactivation of the Ras-ERK1/2-MEK1/2 pathway [105].

The fact that the effects of hydrogen on signaling pathways are not always unidirectional is further confirmed in the study of Wang et al. [61], where hydrogen exerted neuroprotection in a cellular in vitro model of traumatic brain injury through activation of the miR-21/PI3K/Akt/GSK-3β signaling pathway. Activation of the PI3K/Akt kinase pathway has also been found to be related to the neuroprotective effects of molecular hydrogen against neurologic damage and apoptosis in early brain injury induced by subarachnoid hemorrhage [106]. Similarly, protection of mouse hearts against I/R injury by molecular hydrogen involved the activation of the PI3K/Akt pathway [107]. A recent study showed that the Akt kinase pathway plays a critical role in the neuroprotective ability of hydrogen-rich saline (HRS). HRS restored behavioral deficits following hypoxia-ischemia injury in neonatal mice via the activation of Akt kinase pathway [108].

#### 2.3.2. Effects of Molecular Hydrogen on Wnt/β-catenin Signaling

Wnts are secreted glycoproteins. In mammals, the family of Wnt ligands consist of 19 members [109]. Activation of the Wnt signaling system includes canonical and noncanonical pathways which regulate a variety of cellular activities.

Two central components of the canonical Wnt pathway are β-catenin and Axin1. Axin1 plays crucial roles in both the destruction of the β-catenin complex and activation of the LRP6 signaling complex [110]. Wnt ligands in this pathway (Wnt1, Wnt2a, Wnt3a, WNT6, and Wnt8a, WNT9b) bind to the frizzled (FZD) and low-density-lipoprotein-related protein 5/6 (LRP5/6) receptor complex. This complex activates the Dishevelled protein, which, in turn, inhibits the degradation complex that destroys the synthetized β-catenin. The β-catenin destruction complex consists of the central scaffold protein Axin and three other components: (i) tumor-suppressor protein adenomatous polyposis coli (APC), (ii) glycogen synthase kinase-3β (GSK-3β), and (iii) casein kinase-1 (CKI) [111]. Inhibition of the degradation complex leads to the stabilization and translocation of β-catenin into the nucleus. The activation of canonical Wnt signaling depends on the nuclear localization of β-catenin. In the nucleus, β-catenin acts as a transcriptional coactivator which, together with the T-cell factor (TCF) and the lymphoid enhancer factor (LEF) transcription factors, initiates the Wnt transcriptional program [111,112,113]. Potential Wnt/β-catenin downstream target genes are angiogenic factors such as vascular endothelial growth factor (VEGF) [114] and interleukin-8 (IL-8) [115]. Reports have also demonstrated that canonical Wnt signaling can regulate the expression of neurospecific growth factors such as neurotrophin-3 (NT-3) [116] and brain-derived neurotrophic factor (BDNF) [117]. Another possible target of Wnt/β-catenin signaling is the voltage-gated Na^+^ channel NaV1.5 encoded by the SCN5a gene. It was found that the activation of Wnt/β-catenin signaling by hydrogen peroxide inhibited the NaV1.5 expression at the transcriptional level [72].

Canonical Wnt/β-catenin signaling controls cell proliferation and differentiation by regulating the expression of target genes. This cascade plays important roles in the regulation of many cellular functions [112,118,119]. Aberrant activation of this signaling pathway is associated with a number of diseases, including cancers, metabolic, and degenerative diseases [120,121]. Moreover, dysfunction of Wnt signaling has been implicated in age-related diseases [122], and sustained activation of Wnt signaling plays a key role in the pathogenesis of a variety of tissue fibrosis [123]. GSK-3β plays an important role in the destabilization of the Wnt signaling component, β-catenin. Activation of GSK-3β and the disturbed function of Wnt/β-catenin signaling are linked to tissue fibrosis, such as lung fibrosis [124], liver fibrosis [125], and cardiac fibrosis [126].

Lin et al. [55] demonstrated that molecular hydrogen suppressed abnormally activated Wnt/β-catenin signaling by promoting phosphorylation and degradation of β-catenin in cancer L and HeLa cell lines. Hydrogen had no effect on the basal endogenous β-catenin level and inhibited only β-catenin accumulation induced by Wnt3a and GSK3 inhibitors. The protective effects of molecular hydrogen occurred only in situations of aberrant activation of the Wnt/β-catenin pathway, and the data observed using a GSK3 inhibitor pointed to an important role of this kinase in the modulation of β-catenin function. Moreover, the mechanisms of hydrogen-mediated suppression of Wnt/β-catenin signaling did not involve the scavenging of hydroxyl radicals or peroxynitrite. Modulation of the β-catenin pathway by hydrogen was also found in melanocytes [127]. Hydrogen reversed the hydrogen peroxide-induced apoptosis and dysfunction in melanocytes, and the data indicated that the hydrogen-mediated beneficial effects involved Wnt/β-catenin-mediated activation of Nrf2 signaling [127].

Noncanonical Wnt signaling is activated by stimulation of the Frizzled receptor by noncanonical Wnt ligands, such as Wnt4, Wnt5, and Wnt11 [128,129,130]. The activation is β-catenin-independent and may trigger gene transcription by activating a planar cell polarity pathway (PCP) [131,132] and a calcium-dependent pathway (Wnt/Ca^2+)^ [128]. Downstream effectors of the Wnt/PCP pathway may involve the small GTP-binding protein RhoA, c-Jun N-terminal protein kinases (JNK) [133,134], and downstream proteins of the Wnt/Ca^2+^ may involve several kinases, including protein kinase C (PKC) and calcium/calmodulin-dependent kinase (CaMKII) [128,135].

Several studies have shown the effects of molecular hydrogen on components of noncanonical Wnt signaling. It has been documented that a hydrogen-rich medium decreased expression of downstream RhoA effector, Rho-associated coiled-coil protein kinase (ROCK); this was associated with attenuation of LPS-induced vascular hyperpermeability and vascular endothelial-cadherin disruption [136]. Supression of RhoA activity by hydrogen was also found in the human colon cancer cell line, Caco-2; this was connected with hydrogen-mediated amelioration of LPS-induced barrier dysfunction [137]. Zhang et al. [94] described the positive effects of intraperitoneal hydrogen injection on the prevention of isoproterenol-induced cardiac hypertrophy and improvement of cardiac function in mice. Hydrogen exerted its protective effects through blocking several protein kinase pathways (ERK, p38-MAPK), including JNK signaling.

### 2.4. Effects of Hydrogen on Gene Expression Regulation

Hydrogen was found to be involved in regulating the expression of various genes; however, it is not clear whether these regulations are the cause or consequence of the effects of hydrogen against oxidative stress [54]. The primary molecular targets of molecular hydrogen remain unknown, and there is no evidence that hydrogen directly reacts with factors involved in transcriptional regulation. Several lines of evidence indicate that molecular hydrogen can realize its effects in various pathological situations indirectly, through the up- or down- regulation of the expression of distinct genes. The antiapoptotic effects of hydrogen seem to be associated with the upregulation of antiapoptotic factors, and/or the downregulation of proapoptotic factors (Figure 1). It has been observed that hydrogen induced expressions of the antiapoptotic factors Bcl-2 and Bcl-xL [138], and suppressed the expressions of various proapoptotic factors, including caspase 3 [139,140], caspase 8 [138], and caspase 12 [139].

An important factor in modulating responses to oxidative stress, and a major mechanism underlying the cellular protective effects of hydrogen, is the upregulation of the expression of genes that encode several antioxidant enzymes (Figure 1) [7,9,62]. Kawamura et al. [66] reported that inhalation of hydrogen gas during exposure to hyperoxia improved blood oxygenation, reduced inflammation, and induced HO-1 expression in the lung. The HO-1 enzyme participates in the cellular defense against oxidative stress, and its transcription is regulated by nuclear factor erythroid 2-related factor 2 (Nrf2). Nrf-2 seems to play a very important role in mediating the effects of molecular hydrogen on gene expression. Furthermore, Chen et al. [141] reported that molecular hydrogen attenuates the inflammatory response during sepsis by activating the Nrf2-mediated HO-1 signaling pathway.

Hydrogen gas enhances gene expression by promoting the translocation of Nrf2 into the nucleus [142]. A consequence of nuclear Nrf2 translocation is the upregulation of its downstream effectors such as HO-1, SOD, and catalase. The relationship between the effects of molecular hydrogen and Nrf2 was also confirmed by data obtained using a septic model in mice. It was demonstrated that hydrogen therapy protected against intestinal injury induced by oxidative stress and inflammation, and increased the survival rate in WT septic mice, but not in Nrf2-KO mice [143].

Other downstream indirect targets of molecular hydrogen that can be up- or down- regulated include several other genes, such as MMP-2, MMP-9 [144], MMP3, MMP13 [57], cyclooxygenase-2 [145], and connexins [146]. In addition, it has been proposed that molecular hydrogen may affect the expression of several protein kinase signaling pathways [93,147]. However, these molecules are likely downstream and indirectly regulated by H_2_, as the direct targets of H_2_ have yet to be elucidated [54].

### 2.5. Molecular Hydrogen and Autophagy

Autophagy is an adaptive response of cells during conditions of cellular stress, such as nutrient limitation, increased production of ROS, accumulation of protein aggregates or damaged organelles, or presence of extracellular pathogens [148,149]. Through autophagy, the machinery and cytosolic components that are damaged or dysfunctional, along with extracellular pathogens, are degraded via the autophagosome, which fuses with lysosomes to degrade and recycle the sequestered substrates. There are at least three types of autophagy: macroautophagy, microautophagy, and chaperone-mediated autophagy. Basal autophagy helps cells survive by reducing stress sources, and is beneficial to keeping normal cellular functions during the early stage of disease [150,151]. However, excessive autophagy leads to autophagic cell death and is implicated in disease progression [152]. The importance of autophagy for cellular function is emphasized by the fact that autophagy dysfunction can result in impaired mitochondrial function, ROS accumulation, and oxidative stress [153]. A consequence of excessive ROS production could be glutathione depletion and stress of the ER. Functional autophagy plays an important role in protecting cells against death induced by ER stress [151], and its activation may attenuate the ER stress-mediated development of injury. This is also supported by the results of a study showing the protection of the brain from an ischemic insult associated with increased autophagy and reduction of ER stress [150]. Autophagy and ER stress are associated with several pathological conditions, such as neurodegeneration [154], diabetes [155], and hypoxia [156]. Increasing evidence indicates that autophagy may share common molecular inducers and regulatory mechanisms with apoptosis, and a switch in cellular responses from apoptosis to autophagy can lead to cell survival. For example, beclin-1, a protein with a central role in autophagy, may interact with antiapoptotic Bcl-2 family members [157].

Several studies have documented the relationship between hydrogen and autophagy [158,159]. However, consensus regarding the precise role of molecular hydrogen in autophagy regulation has not been reached. Some studies have shown that hydrogen can downregulate autophagy [39,158,159], while others have demonstrated that it induces autophagy [3]. Treatment with hydrogen significantly attenuated neuronal injury in the hippocampal *cornu ammonis* 1 sector. This was associated with autophagy inhibition, and, as a consequence, there was a reduction of brain edema following 24 h of reperfusion [158]. Similarly, hydrogen-mediated downregulation of autophagy was also demonstrated in cardiac cells [159]. In this case, pretreatment with a hydrogen-rich medium suppressed isoproterenol-induced, excessive autophagy in H9c2 cardiomyocytes [159]. In addition, the authors used a mouse model of cardiac hypertrophy and demonstrated that intraperitoneal administration of hydrogen significantly blocked β adrenoceptor agonist-mediated, excessive autophagy in vivo [159]. In contrast to these studies demonstrating the downregulation of autophagy by hydrogen, other studies have documented the opposite effects. It was found that the application of HRS exerted neuroprotection against hypoxic-ischemic brain damage in neonatal mice, and that these effects were mediated in part by the upregulation of autophagy machinery and the downregulation of ER stress [3]. In these cases, the effects of HRS on autophagy pathways included increased LC3B and Beclin-1 expression, and decreased phosphorylation of mTOR and STAT3. Moreover, these changes were associated with the phosphorylation of extracellular-signal regulated protein kinases. Similarly, it was reported that an important role mediating the neuroprotective effects of hydrogen-rich water in a rat model of vascular dementia was the stimulation of FoxO1-mediated autophagy [160]. Additionally, activation of p53-mediated autophagy was shown to play a positive role in HRS-mediated attenuation of acute kidney injury after liver transplantation [161].

Wang et al. [162] found that in a neuropathic pain model, the potent analgesic effects of HRS were associated with the activation of cell autophagy via inducing hypoxia-inducible factor-1α (HIF-1α). This pathway plays a pivotal role in regulating gene expression to maintain oxygen homeostasis [163], and in initiating the transcription of target genes that are important for cellular adaption to hypoxic stimuli. Hypoxia influences autophagy in part via the activation of the HIF1α -dependent pathways [164] (Figure 2).

### 2.6. Molecular Hydrogen and Matrix Metalloproteinases

The extracellular matrix (ECM) plays an important role in the development of cardiac fibrosis, which is also one of the typical clinical features of the aged heart [165]. This process is associated with the modulation of ECM, changes in the gene expression of cardiac fibroblasts, and stimulation of proliferation and phenotypic differentiation of myofibroblasts into fibroblasts [166]. A consequence of cardiac fibroblasts activation is the expression of contractile proteins and the secretion of ECM components, which influence the development of pathophysiological processes [167]. Matrix metalloproteinases (MMPs) play an important role in the regulation and modulation of the ECM. Importantly, it has been documented that molecular hydrogen can regulate the expression of several MMPs, including MMP-2, MMP-9, MMP-3, and MMP-13 [57,144]. Other data suggest that hydrogen-rich water inhibits the migration of smooth muscle cells into vein grafts; this is at least partially explained by lowering the expression of MMP-2 and MMP-9 [168]. MMP-9 was found to promote hemorrhagic infarction by disrupting cerebral vessels in a rat model of middle cerebral artery occlusion (MCAO), but inhalation of molecular hydrogen significantly reduced the MMP-9 expression [144,169]. ROS/RNS play an important mechanistic role in regulating MMP activities/expression, especially peroxynitrite (ONOO–). Therefore, the fact that molecular hydrogen reduces peroxynitrite and other toxic ROS will contribute to the favorable regulation of MMPs.

## 3. Conclusions

The therapeutic potential of molecular hydrogen in the treatment of various diseases may be attributed to several molecular mechanisms. Current information indicates that the cellular protective effects mediated by molecular hydrogen are attributable to the modulation of cellular antioxidant defenses (antioxidant and cytoprotective genes), including intracellular and extracellular redox signaling.

However, the effects of hydrogen on signaling pathways and adaptive cellular responses (e.g., autophagy) are not always unidirectional; both stimulatory and inhibitory effects of molecular hydrogen have been demonstrated.

Further studies are essential to understand the details underlying the regulatory function of molecular hydrogen and the precise mechanisms by which it affects cellular functions in pathological conditions.

## Figures and Tables

**Figure 1 antioxidants-09-01281-f001:**
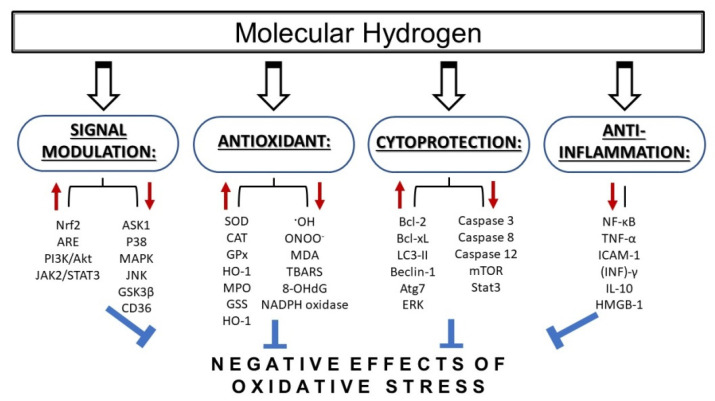
Mechanisms of action of molecular hydrogen in conditions of increased oxidative stress. Molecular hydrogen has been shown to provide protective effects via several mechanisms including antioxidant, anti-inflammatory, and cytoprotective action, as well as via signal modulation. MDA—malondialdehyde; TBARS—thiobarbituric acid reactive substances; 8-OHdG—8-hydroxy-desoxyguanosine; SOD—superoxide dismutase; CAT—catalase; GPx—glutathione peroxidase; HO-1—heme oxygenase 1; MPO—myeloperoxidase; GSS—glutathione synthetase; ASK1—apoptotic signal-regulated kinase 1; MAPK—mitogen-activated protein kinase; JNK—c-Jun N-terminal kinase; CD36—cyclin-dependent kinase 36; Nrf2—nuclear factor-erythroid-2-related factor 2; ARE—antioxidant response elements; NF-κB—nuclear factor kappa B; TNF-α—tumor necrosis factor alpha; ICAM-1—intercellular cell adhesion molecule-1; (INF)γ—interferon gamma; IL-1β—interleukin beta; HMGB-1—high-mobility group box protein 1; mTOR—mammalian target of rapamycin; Stat3—signal transducer and activator of transcription 3; LC3-II—microtubule-associated protein 1A/1B-light chain 3; ERK—extracellular signal-regulated kinase; Atg7—autophagy related 7.

**Figure 2 antioxidants-09-01281-f002:**
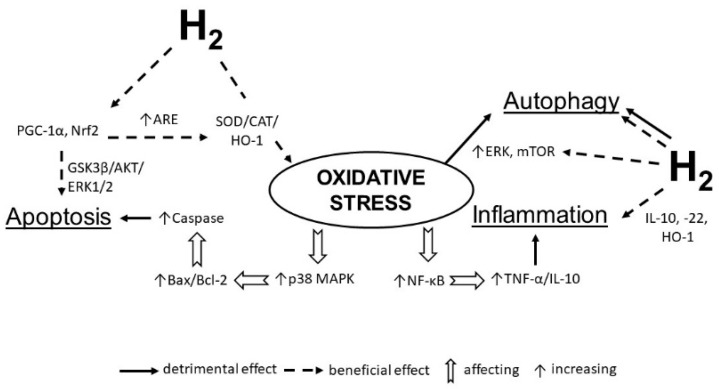
Molecular hydrogen alleviates the negative impact of oxidative stress on the heart. Hydrogen increases the expression of antioxidant enzymes (SOD, catalase, etc.), anti-inflammatory molecules (IL-10, IL-22, HO-1), activates PGC-1α and Nrf2 transcription to decreases apoptosis and regulate autophagy by influencing ERK and mTOR. SOD—superoxide dismutase; IL—interleukin; NF-κB—nuclear factor kappa B; TNF-α—tumor necrosis factor alpha; HO-1—heme oxygenase 1; ERK—extracellular signal-regulated kinase; mTOR—mammalian target of rapamycin; PGC-1α—peroxisome proliferator-activated receptor-gamma coactivator–1 alpha.

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
