# Peer review of "Molecular and Cellular Mechanisms Associated with Effects of Molecular Hydrogen in Cardiovascular and Central Nervous Systems"

_antioxidants, 2020, doi:10.3390/antiox9121281_

Round 1

Reviewer 1 Report

The manuscript submitted by Barancik et al. aims to review the literature regarding the molecular and cellular mechanisms of molecular hydrogen in association with its use in the treatment of disease. The structure of the review is somewhat erratic and does not go into sufficient detail to reflect the title. It is strongly recommended that the pathways and mechanisms involved in H2 protection should be described in more detail, such that the naïve reader can inform him/herself sufficiently. For example, only the canonical Wnt/beta catenin signaling pathway is mentioned. There is no detailed information on the regulation of cytosolic beta-catenin levels (destruction complex), Wnt receptors, TCF4 transcription factor and naming of target genes. Furthermore, it is not mentioned that there are also non-canonical beta-catenin signaling pathways.

General comments

It is not understood why only cardiovascular disease and the central nervous system are described in the review article. Molecular hydrogen appears to also impact diseases of the airways and digestive tract as well as inflammation/sepsis/immune system. The title should be altered to reflect this focus.

Section on “Molecular and Cellular Systems…” should appear before the disease states. The reader can then understand the molecular mechanisms being referred to in the text in Sections 1.2 and 1.3.

The sequence of pathway mentioning and explanation is sometimes incorrect. For example, Nrf2 is mentioned in Section 2.1. but only described in Section 2.3. Please rectify the sequence so that pathways are described in more detail at first mention.

A significant omission is the effect of H2 on mitochondria, especially as mitochondria are the major cellular source of ROS. There are some recent interesting studies from this field, such as Iuchi K et al. Can J Physiol Pharmacol 2019 and Ishihara G, et al. Biophys Biochem Res Commun 2020. This would be timely to include these latest findings.

Specific comments

A pictorial depiction of the chemistry of molecular hydrogen in its antioxidant effect would be desirable.

The arrows in Figure 2 are somewhat confusing. For example, the arrow between “H2” and “oxidative stress” is a positive effect according to the Legend. This is incorrect based on the review article. Possibly the authors meant a positive effect on the antioxidative enzymes. Please improve clarity of the Figure.

An effect of H2 on MMPs is described in Section 2.5 with a sole focus on cardiovascular disease. What about other diseases where MMPs play a central role in the pathophysiology? For example, rheumatoid arthritis (there are a few reports detailing H2 in alleviating the symptoms).

Author Response

Response to reviewers’ comments

Reviewer: 1

Comments to the Author

The manuscript submitted by Barancik et al. aims to review the literature regarding the molecular and cellular mechanisms of molecular hydrogen in association with its use in the treatment of disease. The structure of the review is somewhat erratic and does not go into sufficient detail to reflect the title. It is strongly recommended that the pathways and mechanisms involved in H2 protection should be described in more detail, such that the naïve reader can inform him/herself sufficiently. For example, only the canonical Wnt/beta catenin signaling pathway is mentioned. There is no detailed information on the regulation of cytosolic beta-catenin levels (destruction complex), Wnt receptors, TCF4 transcription factor and naming of target genes. Furthermore, it is not mentioned that there are also non-canonical beta-catenin signaling pathways.

RESPONSE: We thank a lot to reviewer for his/her review of our manuscript and for his/her comments that will improve the quality of our manuscript. We gratefully accept all proposed changes and comments. In the manuscript we added more detailed information about pathways mentioned above and we have also changed the title of manuscript to be more concise.

Comments

It is not understood why only cardiovascular disease and the central nervous system are described in the review article. Molecular hydrogen appears to also impact diseases of the airways and digestive tract as well as inflammation/sepsis/immune system. The title should be altered to reflect this focus.

RESPONSE: We accepted the reviewer’s comment and the title was modified according to recommendation.

Section on “Molecular and Cellular Systems…” should appear before the disease states. The reader can then understand the molecular mechanisms being referred to in the text in Sections 1.2 and 1.3.

RESPONSE: We appreciate your comment. In fact, we have already considered the proposed order when writing the work. However, we feel that it is more natural and attractive for the reader to first read about the pluripotent effects of hydrogen on various diseases, and only then explain the probable mechanism of its action.

A significant omission is the effect of H2 on mitochondria, especially as mitochondria are the major cellular source of ROS. There are some recent interesting studies from this field, such as Iuchi K et al. Can J Physiol Pharmacol 2019 and Ishihara G, et al. Biophys Biochem Res Commun 2020. This would be timely to include these latest findings.

RESPONSE: We accepted the reviewer’s comment and the part about mitochondria was added to the manuscript using suggested literature.

Specific comments

A pictorial depiction of the chemistry of molecular hydrogen in its antioxidant effect would be desirable.

The chemistry of molecular hydrogen and its antioxidant effect has been added.

The arrows in Figure 2 are somewhat confusing. For example, the arrow between “H2” and “oxidative stress” is a positive effect according to the Legend. This is incorrect based on the review article. Possibly the authors meant a positive effect on the antioxidative enzymes. Please improve clarity of the Figure.

RESPONSE: We accepted suggested change and Figure 2 was corrected according the recommendation.

An effect of H2 on MMPs is described in Section 2.5 with a sole focus on cardiovascular disease. What about other diseases where MMPs play a central role in the pathophysiology? For example, rheumatoid arthritis (there are a few reports detailing H2 in alleviating the symptoms).

RESPONSE: The title of the manuscript has been altered. Now it sharpened the focus on cardiovascular and neurological diseases.

Reviewer 2 Report

The beneficial role of molecular hydrogen (H2) as therapeutic in oxidative stress related diseases is an emerging an interesting issue. Although the molecular targets of H2 are not known yet, it seems that its beneficial effects are related mainly with the decrease of oxidative stress.

In this manuscript, Barancik et al. have revised this interesting subject analyzing possible cellular and molecular mechanisms that could be responsible for H2 effects.  The manuscript is well written, and it deserves consideration. I just raise some minor comments that I hope will be useful for the authors:

  1. Line 83: please state the meaning of the abbreviature “ST”.
  2. Line 86: please state the meaning of the abbreviature “LV”.
  3. Sometimes the same pieces of information are repeated several times through the text. For instance, the ability of H2 to reduce Hydroxy radical and peroxynitrite appears at least three times: line 159, 1725, 182. Perhaps, revising the text to avoid unnecessary repetitions might be useful.

Author Response

Response to reviewers’ comments

 Reviewer: 2

Comments to the Author

The beneficial role of molecular hydrogen (H2) as therapeutic in oxidative stress related diseases is an emerging an interesting issue. Although the molecular targets of H2 are not known yet, it seems that its beneficial effects are related mainly with the decrease of oxidative stress. In this manuscript, Barancik et al. have revised this interesting subject analyzing possible cellular and molecular mechanisms that could be responsible for H2 effects.  The manuscript is well written, and it deserves consideration. I just raise some minor comments that I hope will be useful for the authors:

RESPONSE: We thank to reviewer for his nice words and for reviewing our manuscript. His comments will improve the quality of our manuscript. We gratefully accept all proposed changes and comments.

Line 83: please state the meaning of the abbreviature “ST”.

RESPONSE: We accepted the reviewer’s comment and abbreviation was explained.

Line 86: please state the meaning of the abbreviature “LV”.

RESPONSE: We accepted the reviewer’s comment and abbreviation was explained.

Sometimes the same pieces of information are repeated several times through the text. For instance, the ability of H2 to reduce Hydroxy radical and peroxynitrite appears at least three times: line 159, 1725, 182. Perhaps, revising the text to avoid unnecessary repetitions might be useful.

RESPONSE: The whole manuscript was carefully checked and the unnecessary repetitions were avoided.

Reviewer 3 Report

This manuscript it a well written review of molecular hydrogen impact on cell's processes. However, there are few minor points that could be taken into account:

  • last sentence in the abstract could be more direct conclusion based on the data described in text
  • both figures nicely show mechanisms of hydrogen action and its impact on oxidative stress, but it is not clear if this are summarized data gathered from all literature, or those are facts known for human, animal models or mechanisms researched on cell lines, and therefore (un)applicable for understanding medical effects (if data are from research gained on human samples, gene/protein names should be written according to guidelines for human genome and proteom)
  • in Introduction it is stated that „Molecular hydrogen is currently recognized as an emerging therapeutic as its supplemental application exerts protective effects in cardiovascular diseases, neurodegenerative diseases, inflammatory diseases, neuromuscular disorders, metabolic syndrome, diabetes, kidney disorders and cancer.“ But following chapters describe data only for cardiovascular system and CNS – it would be good to add a chapter about other mentioned diseases
  • chapter 2. 3. should be named Effects on hydrogen on gene expression regulation
  • it would be advisable not to use abbreviations in titles and subtitles
  • lines 71, 73 and 383 have some explanations in brackets that just repeat part of the sentence in front
  • there are some redundant spaces thorough text (e. g. line 29, title 2.2.1., …)

Author Response

Response to reviewers’ comments

Reviewer: 3

This manuscript it a well written review of molecular hydrogen impact on cell's processes. However, there are few minor points that could be taken into account:

RESPONSE: We thank to reviewer for reviewing our manuscript and for his comments that will improve the quality of our manuscript. We gratefully accept all proposed changes and comments.

last sentence in the abstract could be more direct conclusion based on the data described in text

RESPONSE: We accepted the reviewer´s comment and the last sentence in the abstract was improved.

both figures nicely show mechanisms of hydrogen action and its impact on oxidative stress, but it is not clear if this are summarized data gathered from all literature, or those are facts known for human, animal models or mechanisms researched on cell lines, and therefore (un)applicable for understanding medical effects (if data are from research gained on human samples, gene/protein names should be written according to guidelines for human genome and proteom)

RESPONSE: The images were processed from the accumulated and generalized of our own data as well as literature data to demonstrate the expected effect of hydrogen in the body. Listing detailed sources would be disruptive to the image.

in Introduction it is stated that „Molecular hydrogen is currently recognized as an emerging therapeutic as its supplemental application exerts protective effects in cardiovascular diseases, neurodegenerative diseases, inflammatory diseases, neuromuscular disorders, metabolic syndrome, diabetes, kidney disorders and cancer.“ But following chapters describe data only for cardiovascular system and CNS – it would be good to add a chapter about other mentioned diseases

RESPONSE: We changed the title of the entire manuscript, which only focus on CVD and CNS. In the present sentence, we want to show pluripotent effect of molecular hydrogen. Writing chapters on other hydrogen diseases and applications would go far beyond the allowable range of manuscript. The work of other authors deals with the other diseases mentioned.

chapter 2. 3. should be named Effects on hydrogen on gene expression regulation

RESPONSE: We accepted the reviewer´s comment and the chapter 2.3 was renamed according recommendation.

it would be advisable not to use abbreviations in titles and subtitles

RESPONSE: We accepted the reviewer´s comment and all abbreviations in the titles were removed.

lines 71, 73 and 383 have some explanations in brackets that just repeat part of the sentence in front

RESPONSE: All explanations in brackets were removed from the manuscript according reviewer´s advice.

there are some redundant spaces thorough text (e. g. line 29, title 2.2.1., …)

RESPONSE: All redundant spaces were removed from the text according reviewer´s suggestion.

Round 2

Reviewer 1 Report

The manuscript by Barancik et al. has been much improved by the authors’ efforts. With the focus on cardiovascular and central nervous systems, the references to other disease states should be removed, such as the addition of lines 496-500. There are still some flaws that hampers the understanding of some concepts, which would be disadvantageous for the reader.

Lines 81-99: The impact of this paragraph is lost by the details of the clinical trials. It is also not understandable why these few clinical trial studies were selected. Why not the report by Sakai T et al. (Vasc Health Risk Manag. 2014) regarding vascular endothelial function or by Yoritaka A et al. (Mov Disord 2013) regarding Parkinson’s disease? Both would be relevant for the review’s foci of cardiovascular and CNS. I feel this paragraph would benefit from widening the literature sources and reducing the experimental conditions of the trials.

In section 1.2, the authors jump into details about cardiovascular diseases and their improvement with molecular H2. There are many unexplained medical terms, which would lose the naïve reader who is not well versed in diseases of the cardiovascular system. It would thus be helpful to include a paragraph on the relevant cardiovascular diseases and their underlying mechanisms, analogous to the CNS section.

Section 2.2 is an important addition to the review. It would benefit from more detailed explanations and there are some sentences wherein the concepts cannot be followed, for example, lines 249-252, I presume the authors mean that hydrogen reduces electron flow to prevent ROS formation. The MMP will be changed as a result but wouldn’t a reduction in MMP also be damaging for the mitochondria? This often occurs during apoptosis. Further, the reference on complex I is missing (line 248) and the focus should be on cardiovascular and CNS.

Minor

Line 46: Please change to endo/sarcoplasmic reticulum (neurons do not have sarcoplasmic reticulum)

Lines 44-48, 69-71, 243-244 are repetitions of each other.

Line 333: remove “which”.

Author Response

Response to reviewers’ comments

Comments to the Author

The manuscript by Barancik et al. has been much improved by the authors’ efforts. With the focus on cardiovascular and central nervous systems, the references to other disease states should be removed, such as the addition of lines 496-500. There are still some flaws that hampers the understanding of some concepts, which would be disadvantageous for the reader.

RESPONSE: We thank a lot to reviewer for review of our manuscript and comments that will improve the quality of our manuscript. We gratefully accept proposed changes and comments.

In revised manuscript we removed part focused on MMP and rheumatoid arthritis (previously lines 496-500). We also removed text and reference which was related to PI3K/Akt and mouse hepatic injury (reference 88) (lines 370-371) 

Lines 81-99: The impact of this paragraph is lost by the details of the clinical trials. It is also not understandable why these few clinical trial studies were selected. Why not the report by Sakai T et al. (Vasc Health Risk Manag. 2014) regarding vascular endothelial function or by Yoritaka A et al. (Mov Disord 2013) regarding Parkinson’s disease? Both would be relevant for the review’s foci of cardiovascular and CNS. I feel this paragraph would benefit from widening the literature sources and reducing the experimental conditions of the trials.

RESPONSE: We thank for your comment. We modified the paragraph focused on clinical trials (lines 84-109).

We reduced the description of experimental conditions of the trials and used suggested literature (reports of Sakai T et al. 2014 and Yoritaka et al. 2013).

 In section 1.2, the authors jump into details about cardiovascular diseases and their improvement with molecular H2. There are many unexplained medical terms, which would lose the naïve reader who is not well versed in diseases of the cardiovascular system. It would thus be helpful to include a paragraph on the relevant cardiovascular diseases and their underlying mechanisms, analogous to the CNS section.

RESPONSE: We appreciate your comment. According to your recommendation we included into section 1.2 a paragraph on the relevant cardiovascular diseases and their underlying mechanisms (lines 113-131 and 145-149).

Section 2.2 is an important addition to the review. It would benefit from more detailed explanations and there are some sentences wherein the concepts cannot be followed, for example, lines 249-252, I presume the authors mean that hydrogen reduces electron flow to prevent ROS formation. The MMP will be changed as a result but wouldn’t a reduction in MMP also be damaging for the mitochondria? This often occurs during apoptosis. Further, the reference on complex I is missing (line 248) and the focus should be on cardiovascular and CNS.

RESPONSE: We thank for your comment. We modified the sentences and content of paragraph focused on potential mechanisms involved in molecular hydrogen effects on mitochondrial function (lines 279-298).

Another paragraphs are in section 2.2 “Molecular hydrogen and mitochondria” focused on cardiovascular diseases and diseases of central nervous system (lines 304-325).

We added missing reference (Line 279, Ishihara et al. 2000).  

Minor comments

Line 46: Please change to endo/sarcoplasmic reticulum (neurons do not have sarcoplasmic reticulum)

RESPONSE: We accepted suggested change and corrected text according the recommendation.

 Lines 44-48, 69-71, 243-244 are repetitions of each other.

RESPONSE: We accepted reviewers’ recommendation and all problematic sentences were corrected.

 Line 333: remove “which”.

RESPONSE: We removed the word “which”.